# Sex Differences in Clinical Course and Intensive Care Unit Admission in a National Cohort of Hospitalized Patients with COVID-19

**DOI:** 10.3390/jcm10214954

**Published:** 2021-10-26

**Authors:** Irit Nachtigall, Marzia Bonsignore, Petra Thürmann, Sven Hohenstein, Katarzyna Jóźwiak, Michael Hauptmann, Sandra Eifert, Julius Dengler, Andreas Bollmann, Heinrich V. Groesdonk, Ralf Kuhlen, Andreas Meier-Hellmann

**Affiliations:** 1Department of Infectious Diseases and Infection Prevention, Helios Hospital Emil-von-Behring, 14165 Berlin, Germany; 2Institute of Hygiene and Environmental Medicine, Charité—Universitätsmedizin Berlin, 12203 Berlin, Germany; 3Center for Hygiene, Evangelische Kliniken Gelsenkirchen, 45879 Gelsenkirchen, Germany; marziabonsignore@gmx.de; 4Philipp Klee-Institute for Clinical Pharmacology, Helios University Hospital Wuppertal, 42283 Wuppertal, Germany; petra.thuermann@helios-gesundheit.de; 5Department of Clinical Pharmacology, University Witten Herdecke Faculty of Health Witten, 58455 Witten, Germany; 6Department of Electrophysiology, Heart Center Leipzig at University of Leipzig, 04289 Leipzig, Germany; Sven.Hohenstein@helios-gesundheit.de (S.H.); Sandra.Eifert@helios-gesundheit.de (S.E.); Andreas.Bollmann@helios-gesundheit.de (A.B.); 7Leipzig Heart Digital at Leipzig Heart Institute, 04289 Leipzig, Germany; 8Institute of Biostatistics and Registry Research, Faculty of Health Sciences Brandenburg, Brandenburg Medical School Theodor Fontane, 16816 Neuruppin, Germany; katarzyna.jozwiak@mhb-fontane.de (K.J.); michael.hauptmann@mhb-fontane.de (M.H.); 9Brandenburg Medical School Theodor Fontane, Campus Bad Saarow, 15526 Bad Saarow, Germany; julius.Dengler@helios-gesundheit.de; 10Department of Neurosurgery, Helios Hospital Bad Saarow, 15526 Bad Saarow, Germany; 11Department of Interdisciplinary Intensive and Intermediate Care, Helios Hospital Erfurt, 99089 Erfurt, Germany; Heinrich.Groesdonk@helios-gesundheit.de; 12Helios Health GmbH, 10117 Berlin, Germany; Ralf.kuhlen@helios-health.com; 13Helios Kliniken GmbH, 10117 Berlin, Germany; andreas.Meier-Hellmann@helios-gesundheit.de

**Keywords:** sex differences, age dependency, intensive care unit, COVID-19, mechanical ventilation

## Abstract

Males have a higher risk for an adverse outcome of COVID-19. The aim of the study was to analyze sex differences in the clinical course with focus on patients who received intensive care. Research was conducted as an observational retrospective cohort study. A group of 23,235 patients from 83 hospitals with PCR-confirmed infection with SARS-CoV-2 between 4 February 2020 and 22 March 2021 were included. Data on symptoms were retrieved from a separate registry, which served as a routine infection control system. Males accounted for 51.4% of all included patients. Males received more intensive care (ratio OR = 1.61, 95% CI = 1.51–1.71) and mechanical ventilation (invasive or noninvasive, OR = 1.87, 95% CI = 1.73–2.01). A model for the prediction of mortality showed that until the age 60 y, mortality increased with age with no substantial difference between sexes. After 60 y, the risk of death increased more in males than in females. At 90 y, females had a predicted mortality risk of 31%, corresponding to males of 84 y. In the intensive care unit (ICU) cohort, females of 90 y had a mortality risk of 46%, equivalent to males of 72 y. Seventy-five percent of males over 90 died, but only 46% of females of the same age. In conclusion, the sex gap was most evident among the oldest in the ICU. Understanding sex-determined differences in COVID-19 can be useful to facilitate individualized treatments.

## 1. Introduction

Since the beginning of the coronavirus disease (COVID-19) pandemic, there has been increasing evidence that males are at higher risk for a severe course or death than females despite similar SARS-CoV-2 infection rates [1,2,3]. Several studies have identified male gender and older age [4,5,6] as risk factors for intensive care unit (ICU) admission and death. Hormonal [7], genetic [8] and gender-specific differences in the immune response [9] have been discussed as possible causes for the different results in COVID-19. It has been shown that there is a stronger inflammatory response in men, which is likely to contribute to the severity of the disease [10]. The identification of the causes of sex-specific differences among COVID-19 patients is important in order to guide medical decision making, e.g., by development of new therapeutic targets, or to use healthcare resources efficiently. We believe that a more in-depth understanding of sex-specific differences during the different steps of the clinical course of hospitalized COVID-19 patients will help to disentangle the role of gender and other prognostic clinical factors. Here, we describe the age- and sex-specific differences in clinical determinants of the course of hospitalized COVID-19 patients in a large cohort and how these differ between patients on intensive care units (ICU), patients requiring invasive or noninvasive mechanical ventilation and patients stable enough to be treated in a normal ward.

## 2. Materials and Methods

### 2.1. Study Design

The research was conducted as an observational retrospective cohort study. All patients hospitalized with the ICD-10 code U07.1. in each of the 83 hospitals of the Helios Group between 4 February 2020 and 22 March 2021 were included. The ICD-10 code U07.1 is defined as PCR-confirmed infection with SARS-CoV-2. Helios is a privately owned company with hospitals spread throughout Germany. The proportion of basic to tertiary care is representative for the overall distribution of hospitals in Germany. The patient mix is representative, since all Helios hospitals are fully covered by all health care insurance plans. Information on sex, age, length of stay, ICU, mechanical ventilation, comorbidities and death was retrieved from claims data. Mechanical ventilation was defined as ventilation with pressure support via either invasive devices like tracheal tube or tracheostomy, or use of noninvasive devices. Mortality was defined as death during the same hospital stay. Claims data on comorbidity were summarized in the Elixhauser Comorbidity Index [11] for further analysis. The Elixhauser Comorbidity Index is a method of categorizing patient comorbidities based on the International Classification of Diseases (ICD) diagnostic codes in administrative data. In Table 1, phenotypes and their respective weights for the calculation are shown.

Data on symptoms were collected from the medical records by trained infection control nurses and entered into our hospitals’ routine infection control system; this process has been described elsewhere [12]. Data on COVID-19 symptoms at admission were not available for all patients (85.3%) since they were derived from a manually managed database. For the analysis of in-hospital mortality, we excluded cases with discharge due to hospital transfer or unspecified reasons (8.1% of all patients). Claims data and data from the infection control system were linked by a pseudonymised hospital case number.

### 2.2. Statistical Analysis

We defined three cohorts: patients treated on normal wards only (non-ICU patients), patients treated in intensive care unit at any time during their hospital stay (ICU patients) and the subgroup of ICU-patients who were mechanically ventilated. For the description of the patient characteristics of the cohorts, we employed χ2-tests for binary variables and analysis of variance for numeric variables in the R environment for statistical computing (version 4.0.2, 64-bit build, R Core Team. 2020. R. A Language and Environment for Statistical Computing; https://www.R-project.org/; accessed on 20 October 2021). We report proportions, means, standard deviations, and *p*-values. For the comparison of proportions of symptoms, as well as selected treatments and outcomes in the different cohorts, we used logistic generalized linear mixed models (GLMMs). We specified hospitals as random factor with varying intercepts. These models take into account the variability of the hospitals with respect to the dependent variables. Observations of institutions with very low or high values were shrunken towards the mean, and hence the corresponding cases entered the analyses with lower weight. The analysis of the outcome variable length of stay was performed via linear mixed models using a log-transformed outcome variable. Multivariable logistic GLMMs were used to assess the influence of sex on the risk of admission to the ICU and death. These models included the predictors sex, age, and their interaction, as well as the Elixhauser comorbidity index, and they allowed for inferring the adjusted impact of sex. To predict mortality at admission to ICU or a normal ward, we developed a model based on parameters easily available at that time via logistic regression based on sex, age, and the interaction of these variables, with age as a linear and quadratic predictor (second-grade polynomial). Compared to a model with age as a linear predictor, the reported model proved to be superior in terms of Akaike (AIC) and the Bayesian Information Criterion (BIC). The final model allowed for predicting mortality of females and males based on their age. We used Poisson GLMMs with log function for the analysis of absolute case numbers, and report crude-risk ratios, confidence intervals, and *p*-values.

## 3. Results

A total of 23,235 patients testing positive for SaRS-CoV-2 were admitted to 83 hospitals between 4 February 2020 and 22 March 2021; among them more males than females (51.4% males, crude-risk ratio = 1.06, 95% confidence interval CI = 1.03–1.08). Table 2 provides detailed patient characteristics. Admission to the ICU was documented for 26.3%, and mechanical ventilation for 15.6% of all patients. On average, males were about 2.2 years younger than females (mean ± SD = males 67.9 ± 17.4, females 70.1 ± 19.8, *p*-value < 0.0001). In the total cohort, males had a higher prevalence of comorbidities as measured by the Elixhauser Comorbidity Index (mean ± SD = males 11.7 ± 11.8, females 10.9 ± 11.3, *p*-value < 0.0001). The comorbidity index was higher in ICU than in non-ICU patients (16.0 ± 12.3 vs. 9.6 ± 10.8, *p* < 0.0001) and highest in ventilated ICU patients (17.0 ± 12.3, *p* <0.0001). In contrast to the total cohort, the Elixhauser comorbidity index did not differ significantly between the sexes in the non-ICU/ICU/ventilated ICU subcohorts. The Elixhauser index increased with age in the total cohort (Figure 1).

At hospital admission, more COVID-19 symptoms were documented for male than female patients i.e., fever (34.9% vs. 26.5%; OR = 1.55, 95% CI = 1.45–1.65), cough (26.1% vs. 22.5%, OR = 1.24, 95% CI = 1.16–1.33) or dyspnoea (29.9% vs. 22.9%, OR = 1.47, 95% = 1.37–1.57).

Patients admitted to the ICU at any time during their hospital stay presented more often with fever (34.5% vs. 29.5%; OR = 1.29; 95% CI = 1.20–1.39) or dyspnoea (35.8% vs. 23.1%; OR = 2.02; 95% CI = 1.88–2.18) at admission compared to non-ICU-patients, again with the same male preponderance.

In total, 21.1% of all patients admitted to a hospital with SARS-CoV-2 died, 47.0% of them had been treated in an ICU (Table 3). The risk of death was higher for males than females (OR = 1.31, 95% CI = 1.22–1.40); higher risk for males was also observed among ICU and non-ICU-patients (Table 3). Mortality was highest among ICU patients who had been ventilated (58.4%) compared to all ICU patients (41.1%, *p* < 0.0001) and to non-ICU patients (14.8%, *p* < 0.0001) with no significant difference between sexes in the ventilated group.

In the multivariable analysis of the admission to intensive care and of in hospital mortality, sex and age were independent risk factors (Table 4 and Table 5).

To compare the risk of death according to age, we developed a mortality prediction model for the non-ICU and the ICU cohort (Figure 2). Until the age of around 60, the predicted mortality increased with no substantial difference between sexes. After 60, the risk of death increased to a larger extent in males than in females, resulting in a diverging gap. The gap statistically translates into a significantly positive interaction between age and sex. In non-ICU patients, the predicted mortality of a 75-year (y) old female was around 11% and matched the mortality of a male of 71 y. At 90, a female had a predicted risk of 31%, corresponding to a male of 84 y. In the ICU cohort, this gap was even more divergent among the oldest: the predicted mortality of a 75-years old female was around 44% and matched the mortality of a male of 71 y. At 90, a female had a risk for death of 46%, equivalent to a male of 72 y. While the probability of death steadily increased for males until a maximum of 75% in the oldest, it plateaued for females at the level of ca. 46% after 90 y and decreased in the highest age group. The results in the highest age group, though, had little predictive power due to the low number of very elderly patients.

## 4. Discussion

We evaluated more than 23,000 patients with SARS-CoV-2 infections admitted to 83 hospitals. Male sex, age and the comorbidity score were independent risk factors for ICU admission and death. In our prediction model, mortality increasingly diverged between the sexes with age. This sex gap was greatest among the oldest on ICU, where the predicted mortality reached 46% for females and 75% for males.

In our study, more males were hospitalized with SARS-CoV-2 than females. There has been an ongoing debate as to whether males might be more likely to get infected with SARS-CoV-2 than females due to higher receptivity, possibly caused by comorbidities and a riskier lifestyle. Early studies originating mostly from China indicated a bias towards males [13]. In Germany, however, as of March 2021 the national COVID-19 survey shows a female predominance in reported cases, with a proportion of males of 48% [14]. Most European countries present a similar ratio [4]. This male bias of hospitalized patients in our study from a population with a lower male proportion of infections could result from the more severe course of COVID-19 in males, leading to a higher hospital admission rate.

Male sex and age of patients [4,5,6] have been identified as risk factors for ICU admission and death in several studies. A former study on the effect of sex and age on the clinical course of COVID-19 showed an increased risk of death among younger men [6]. In contrast to this, we found a sex difference only in patients older than 60 years.

Among hormonal and genetic differences, sex-specific differences in the immune response have been discussed as possible causes for the difference in outcomes in COVID-19 infection. Males have been shown to develop a stronger inflammatory response, likely contributing to disease severity [10]. In a further study, SARS-CoV-2 infection induced higher plasma levels of innate immune cytokines and chemokines in males, along with more robust induction of nonclassical monocytes [9]. In contrast, female patients displayed significantly more robust T cell activation. Higher innate immune cytokines in female patients were associated with worse disease progression, but not in male patients. T cell response diminished with increasing age in males and was predictive of worse disease outcome in male patients. In female patients, even older patients were able to develop robust T cell responses [9]. This sex difference in immune response in older age could explain the sex gap, which was found in our dataset to begin after the age of menopause and to further increase with age. The protecting effect of female sex increased in the same years when females experience a decline in their production of sex hormones. Thus, female sex hormones cannot be the supporting element in this advantage. The different immune response may also be the reason why more symptoms were documented among males than females upon admission. Apart from the underlying immunology, sex differences in behaviors, comorbidities, and access to healthcare may explain or contribute to the observed pattern [15].

In our study, females were less likely to receive ICU care. Females admitted to ICU, or mechanically ventilated, were older than males, but had a similar comorbidity index. Earlier studies have identified a shortage of medical care towards females; females had a lower likelihood to receive ICU care than males, even when being more severely ill [16,17,18], which can explain why females have a higher ICU mortality due to sepsis [19]. However, the lower mortality of female ICU-patients in our study does not indicate a shortage of medical need. Intriguingly, once ventilated, females seemed to lose their survival advantage. Presumably, mechanical ventilation represents the final stage of the disease, after the sex specific protective factors have failed.

### Strengths and Limitations

The advantage of our study is the large number of patients derived from the entire Helios network since the onset of the pandemic, allowing for analysis of claims data linked to our routine infection control system. There are several limitations to our study. We could not differentiate infection with SARS-CoV-2 as the actual cause of death from other causes in which the infection was not the cause of death, although present. Due to the type and structure of the analyzed database information regarding patient-specific imaging, laboratory results or medication was not available. Patient characteristics did not include times from symptoms onset to hospital or ICU admission or a severity illness score. The lacking information could have caused multiple biases and should be highlighted in future research. With this descriptive data analysis, we can only speculate on the mechanism of the age-dependent increasing risk gap for unfavorable outcome between the sexes. One explanation might have been a more pronounced age-dependent increase of comorbidities in men. However, we found no significant differences in Elixhauser indices between sexes for increasing age groups. Further research should address possible underlying mechanisms. Our applied treatment strategies should be reconsidered for sex-dependent different efficacies.

## 5. Conclusions

We were able to show sex differences and their age-dependency in the outcomes of COVID-19 for nonintensive and intensive-care patients. The sex gap was most evident among the oldest in the intensive care unit. While males over 90 years had a sharply reduced chance of survival, more than half of females in this age group still survived. To our knowledge, this interaction of sex and age on mortality, especially among ICU and ventilated patients, has not been described before. Understanding sex-determined differences in COVID-19 can be useful to facilitate individualized treatments.

## Figures and Tables

**Figure 1 jcm-10-04954-f001:**
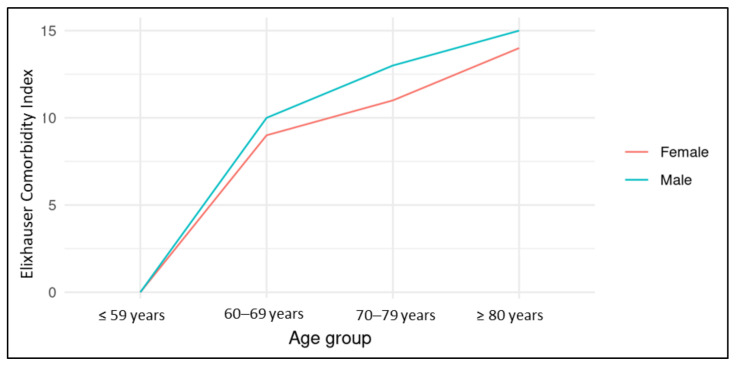
Elixhauser Comorbidity Index (Median) in Dependence of Age. The interaction was not significant (*p* = 0.06).

**Figure 2 jcm-10-04954-f002:**
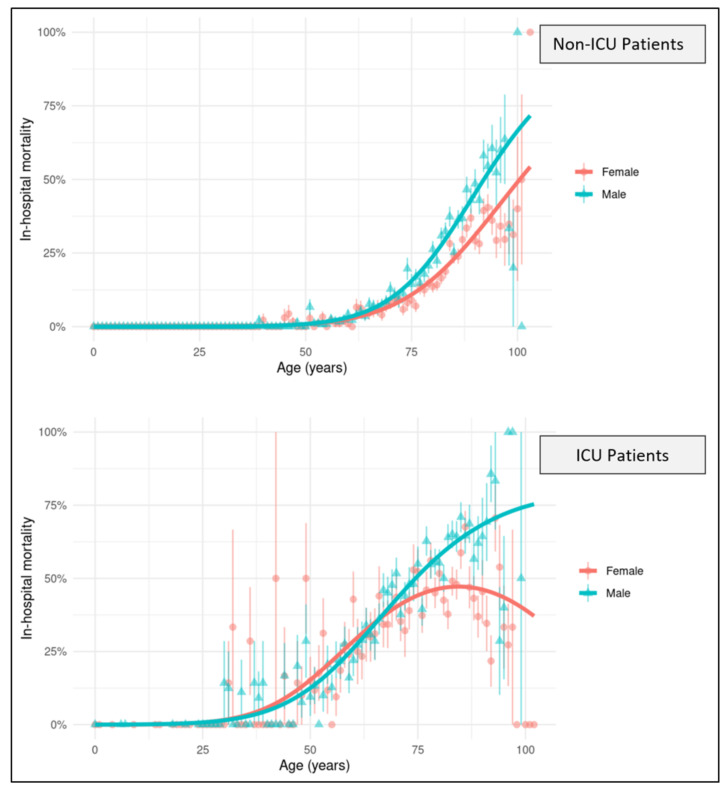
Average In-Hospital Mortality of Males (Circles) and Females (Triangles). Error bars represent standard errors. The curves show the model fit.

**Table 1 jcm-10-04954-t001:** Phenotypes of the Elixhauser Comorbidity Index and Respective Weights for Calculation.

Phenotype	Weight
Congestive heart failure	9
Cardiac arrhythmias	0
Valvular disease	0
Pulmonary circulation disorders	6
Peripheral vascular disorders	3
Hypertension (combined uncomplicated and complicated)	−1
Paralysis	5
Other neurological disorders	5
Chronic pulmonary disease	3
Diabetes, uncomplicated	0
Diabetes, complicated	−3
Hypothyroidism	0
Renal failure	6
Liver disease	4
Peptic ulcer disease, excluding bleeding	0
AIDS/HIV	0
Lymphoma	6
Metastatic cancer	14
Solid tumour without metastasis	7
Rheumatoid arthritis/collagen vascular diseases	0
Coagulopathy	11
Obesity	−5
Weight loss	9
Fluid and electrolyte disorders	11
Blood loss anaemia	−3
Deficiency anaemia	−2
Alcohol abuse	−1
Drug abuse	−7
Psychoses	−5
Depression	−5

**Table 2 jcm-10-04954-t002:** Patient Characteristics by Sex among Non-Intensive-Care (Non-ICU), ICU Patients (Ventilated and Non-ventilated) and Ventilated ICU Patients.

	Non-ICU Patients (*n* 17,133)				ICU Patients (Ventilated and Non-Ventilated) (*n* = 6102)				Ventilated ICU-Patients (*n* = 3654)			
	Females *n* (%)	Males *n* (%)	*p*-Value	Odds Ratio/Crude-Risk Ratio Males (95% CI)	Females *n* (%)	Males *n* (%)	*p*-Value	Odds Ratio/Crude-risk Ratio Males (95% CI)	Females *n* (%)	Males *n* (%)	*p*-Value	Odds Ratio/Crude-Risk Ratio Males (95% CI)
	8884 (51.9)	8249 (48.1)	<0.0001	0.93 (0.90–0.96)	2411 (39.5) (21.3% of females)	3691 (60.5) (30.9% of males)	<0.0001	1.53 (1.45–1.61)	1278 (35.0)	2376 (65.0)	<0.0001	1.86 (1.74–1.99)
**Age (years)**												
Mean (SD)	69.3 ± 20.9	67.3 ± 18.8	<0.0001		72.9 ± 14.9	69.2 ± 13.7	<0.0001		71.1 ± 13.8	69.2 ± 12.4	<0.0001	
≤59	2407 (27.1)	2426 (29.4)	0.0008		417 (17.3)	807 (21.9)	<0.0001		241 (18.0)	483 (20.3)	n.s.	
60−69	953 (10.7)	1406 (17.0)	<0.0001		364 (15.1)	887 (24.0)	<0.0001		236 (18.5)	640 (26.9)	<0.0001	
70−79	1650 (18.6)	1736 (21,0)	<0.0001		634 (26.3)	1034 (28.0)	n.s.		390 (30.5)	694 (29.2)	n.s.	
≥80	3874 (43.6)	2681 (32.5)	<.0001		996 (41.3)	963 (26.1)	<0.0001		411 (32.2)	559 (23.5)	<0.0001	
**Elixhauser Comorbidity Index**												
Mean (SD)	9.5 ±10.6	9.7 ± 11.0	n.s.		15.9 ± 12.4	16.1 ± 12.3	n.s.		16.8 ± 12.4	17.0 ±12.3	n.s.	
**Symptoms at admission ***												
Fever	1953 (25.9)	2352 (33.5)	<0.0001	1.51 (1.40–1.63)	608 (29.0)	1217 (38.1)	<0.0001	1.54 (1.36–1.75)	418 (37.3)	911 (43.4)	0.0002	1.36 (1.16–1.60)
Dyspnea	1561 (20.7)	1807 (25.7)	<0.0001	1.37 (1.26–1.49)	647 (30.8)	1248 (39.0)	<0.0001	1.46 (1.29–1.66)	505 (45.0)	973 (46.3)	n.s.	1.07 (0.91–1.25)
Cough	1700 (22.5)	1844 (26.2)	<0.0001	1.24 (1.13–1.35)	467 (22.2)	820 (25.6)	0.0046	1.22 (1.06–1.39)	306 (27.3)	588 (28.0)	n.s.	1.05 (0.89–1.24)
Diarrhea	619 (8.2)	476 (6.8)	0.0008	0.81 (0.71–0.91)	128 (6.1)	215 (6.7)	n.s.	1.11 (0.88–1.39)	62 (5.5)	158 (7.5)	0.0254	1.41 (1.04–1.91)
**Length of hospital stay (nights)**												
Mean (SD)	10.1 ± 10.6	9.8 ±9.8	n.s.		18.8 ± 17.3	19.4 ± 16.4	0.0238		19.8 ± 18.6	20.7 ± 16.9	0.0052	
**ICU stay (days)**												
Mean (SD)	n.a.	n.a.	n.a.	n.a.	8.0 ± 10.9	10.4 ± 12.2	<0.0001		12.1 ± 13.1	14.1 ± 13.5	<0.0001	

* Symptoms were available for a subset of cases (85.3%). For the analysis of in-hospital mortality, we excluded cases with discharge due to hospital transfer or unspecified reasons (8.1% of all patients). ICU = Intensive Care Unit, 95% CI = 95% confidence interval, SD = standard deviation, n.s.: = not significant, n.a = not applicable.

**Table 3 jcm-10-04954-t003:** Comparison of In-Hospital Mortality.

	Proportion (*n/N*)			Gender		Comparison with Non-ICU		Interaction	
Cohort	Females and Males	Females	Males	Odds Ratio (95% CI)	*p* Value	Odds Ratio (95% CI)	*p* Value	Odds Ratio (95% CI)	*p* Value
Total (*N* = 21,346)	21.1% (4509/21,346)	18.8% (1969/10,488)	23.4% (2540/10,858)	1.31 (1.22–1.40)	<0.0001				
non-ICU (*N* = 16,197)	14.8% (2391/16,197)	13.7% (1152/8399)	15.9% (1239/7798)	1.19 (1.09–1.30)	0.0001				
ICU (*N* = 5149)	41.1% (2118/5149)	39.1% (817/2089)	42.5% (1301/3060)	1.14 (1.02–1.28)	0.0259	4.17 (3.87–4.49)	<0.0001	0.97 (0.84–1.13)	0.7217
ICU and ventilated (*N* = 2959)	58.4% (1728/2959)	59.5% (633/1064)	57.8% (1095/1895)	0.93 (0.80–1.09)	0.3996	8.55 (7.81–9.35)	<0.0001	0.79 (0.67–0.95)	0.0109
ICU and not ventilated (*N* = 2190)	17.8% (390/2190)	18.0% (184/1025)	17.7% (206/1165)	0.98 (0.79–1.22)	0.8487	1.32 (1.17–1.48)	<0.0001	0.83 (0.65–1.05)	0.1124

Based on 21,346 cases (91.9%). We excluded cases with discharge due to hospital transfer or unspecified reason.

**Table 4 jcm-10-04954-t004:** Results of Multivariable Analysis of the Probability of Intensive Care.

Variable	OR (95% CI)	*p*-Value
Male sex	1.588 (1.491–1.692)	<0.0001
Age (years)	0.997 (0.995–0.999)	0.0050
Interaction sex × age	0.996 (0.993–1.000)	0.0319
Elixhauser Comorbidity Index	1.053 (1.050–1.056)	<0.0001

**Table 5 jcm-10-04954-t005:** Results of Multivariable Analysis of In-Hospital Mortality.

Variable	OR (95% CI)	*p*-Value
**Total cohort**		
Male sex	1.415 (1.281–1.563)	<0.0001
Age (years)	1.064 (1.061–1.068)	<0.0001
Interaction sex × age	1.015 (1.008–1.022)	<0.0001
Elixhauser Comorbidity Index	1.051 (1.047–1.054)	<0.0001
**ICU cohort**		
Male sex	1.277 (1.122–1.454)	0.0002
Age (years)	1.053 (1.047–1.058)	<0.0001
Interaction sex × age	1.032 (1.021–1.044)	<0.0001
Elixhauser Comorbidity Index	1.041 (1.035–1.047)	<0.0001
**Non-ICU cohort**		
Male sex	1.312 (1.096–1.570)	0.0031
Age (years)	1.097 (1.091–1.104)	<0.0001
Interaction sex × age	1.016 (1.005–1.027)	0.0053
Elixhauser Comorbidity Index	1.037 (1.032–1.042)	<0.0001

## Data Availability

Helios Health and Helios Hospitals have strict rules regarding data sharing because health data are a sensitive data source and have ethical restrictions imposed due to concerns regarding privacy. Access to anonymized data that support the findings of this study are available on request from the Leipzig Heart Institute (www.leipzig-heart.de; accessed on 24 October 2021). Please direct queries to the data protection officer (Email: info@leipzig-heart.de) and refer to study “eCaRe-COVID19” (HCRI ID 2020-0369).

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
