# Peer review of "Sex Differences in Clinical Course and Intensive Care Unit Admission in a National Cohort of Hospitalized Patients with COVID-19"

_jcm, 2021, doi:10.3390/jcm10214954_

Round 1

Reviewer 1 Report

All my comments and concerns were adequately addressed. I just have a minor comment on the revised manuscript. Some "0" in the table have a different font compared to the other numbers.

Author Response

We would like to thank the reviewers for their suggestions for improvement. We tried to address all the comments and hope that you now find it sufficient for publication. In the following, we will try our best to answer the questions that have been raised.

Changes are highlighted in the manuscript by red color.

Reviewer 1

All my comments and concerns were adequately addressed. I just have a minor comment on the revised manuscript. Some "0" in the table have a different font compared to the other numbers

Thank you for your thoughtful look, we have corrected this accordingly.

Reviewer 2

Authors have addressed most of my comments and the quality of the manuscript has significantly improved. However, there are still minor flaws:

-The numbering of the references is wrong - in the introduction section references ranging from 1 to 9 are given and then suddenly no 14. Please correct it.

Thank you for your thoughtful look, we have corrected this accordingly.

-Font is still inappropriate - Authors must use Palatino Linotype font according to the Journal's requirement

Thank you for your thoughtful look, we have corrected this accordingly

Reviewer 3

Major comments:

1) Elixhauser Comorbidity Index

Authors used Elixhauser Comorbidity Index as a method to assess comorbidities in COVID-19 affected patients (lines 72-73). But in the current form, it was not detailed further in the manuscript. The index calculation should be presented in the text and in a table in order to know which were the included comorbidities and how this index can be interpreted.

We have included now a table with the phenotypes of the Elixhauser index together with the respective weights for the calculation. Please see table 1 in the manuscript.

2) Patients’ characteristics – times between symptoms onset and admission

Patient characteristics did not include times from symptoms onset to hospital admission and to ICU admission, severity illness score or a risk score of clinical deterioration such as NEWS2. For example, if a large proportion of male patients had a longer time between symptoms onset to hospital admission and were finally admitted in hospital with more severe disease, consequently they may have a higher risk of death or ICU. Then, these variables should be considered since it may introduce bias in results interpretation.

The question is indeed very interesting, unfortunately this data is not included in our dataset. We have written this into the limitations.

3) Patients’ characteristics – Different data in the text and in table 1

Data presentation could be clearer since data presented in the text are different from those reported in the table. In the text, authors reported global data on symptoms at admission (lines 119-125), and in-hospital mortality (lines 131-135) in all patients and in the subgroups, but these data are not presented in tables.

For example, “In total, 21.1% of all patients admitted to a hospital with SARS-CoV-2 died, 47.0% of them had been treated on ICU.” (line 131). These data are not reported in table 1 and not easy to check. “21,1%” can be a percentage of 4,509 intra-hospital deaths (1,152+1,239+817+1,301) on a total of 21,353 patients after exclusion of 8.1%, but it is not clearly presented. All the corresponding data should be presented in the manuscript in a new table or in a new table 1 in order to be clear.

Thank you for your remark. To not be redundant, we intentionally provided information in the text that is not included in the table and vice versa. To make it easier to understand the data at a glance, we have divided the table with the basic characteristics and now there is a detailed list of hospital mortality in five different groups: Total, Non-ICU, ICU, ICU with ventilation, ICU without ventilation. For each of these groups, the total mortality is given (% (n/N)), both for both sexes combined and separately. In addition, ORs and p-values of the comparison between the sexes are given in each case. Below the table is an indication of the proportion of the total data used for the mortality analysis.

Please see table 3 in the manuscript

4) Multivariable analysis and independent risk factors for in hospital mortality in ICU.

Authors used multivariable logistic GLMMs to assess the influence of sex on the risk of admission to the ICU and death in the total cohort (tables 2 and 3).

As a mortality prediction model was further applied to the non-ICU and the ICU cohort, it should be interesting to present multivariate analysis of in hospital mortality in these subgroups in order to known whether age, sex, interaction sex / age, Elixhauser comorbidity index are also independent risk factors.

We have made new calculations which can now be found in an extended table 5

Minor comments:

1) Mortality prediction model in ICU cohort

To compare the risk of death according to age, authors developed a mortality prediction model for the Non-ICU and the ICU cohort (Figure 2). In the statistic section, the model predicting in-hospital mortality was constructed with logistic regression based on sex, age, and the interaction of these variables, with age as a linear and quadratic predictor (second-grade polynomial). As table 1 reported that ICU stay was longer in male than in female patients (10.4 vs 8.0; p<.0001) and that, in ICU, the proportion of ventilated patients was higher in male than in female (65.0% vs 35.0%, p<.0001), it should be interesting to known if these potential explanatory variables have been included in the construction of the mortality prediction model adapted for ICU patients.

This is indeed an interesting question that we would like to answer with the following 2 tables. However, our intention was to develop a model that predicts mortality at the beginning of the stay in the intensive care unit or in the normal ward. To do this, we used the parameters that are easily available at the time of admission. Adding ventilation and intensive care stay would make the model no longer usable for prediction at admission, as these are only available during the course. This would make it more of a retrospective analysis than a prediction model.

Therefore, instead of including the tables in the article, we extended the methods sections with further information on the prediction model.

Table 1: Results of multivariable analyses of in-hospital mortality in the ICU cohort

Variable

OR (95% CI)

P value

Male sex

1.12 (0.97-1.29)

0.1357

Age

1.08 (1.07-1.08)

< 0.0001

Interaction sex × age

1.03 (1.01-1.04)

< 0.0001

Elixhauser comorbidity score

1.04 (1.04-1.05)

< 0.0001

Length of stay at ICU

0.99 (0.98-0.99)

< 0.0001

Mechanical ventilation

11.71 (9.85-13.92)

< 0.0001

* Based on 5,149 cases (84.4%). We excluded cases with discharge due to hospital transfer or unspecified reason.

Table 2: Results of multivariable analyses of in-hospital mortality in the non-ICU cohort.

Variable

OR (95% CI)

P value

Male sex

1.30 (1.08-1.56)

0.0052

Age

1.10 (1.09-1.11)

< 0.0001

Interaction sex × age

1.02 (1.01-1.03)

0.0044

Elixhauser comorbidity score

1.04 (1.03-1.04)

< 0.0001

Mechanical ventilation

11.20 (7.78-16.12)

< 0.0001

* Based on 16,197 cases (94.5%). We excluded cases with discharge due to hospital transfer or unspecified reason.

2) Comparison of in-hospital mortality

Authors compared the in-hospital mortality ICU ventilated patients to all ICU patients and non-ICU patients. It should be interesting to compare in-hospital mortality in non-ICU patients to ICU patients, and ventilated ICU patients to non-ventilated ICU patients, and then focus on sex differences among the different groups.

Thank you for this question, which we answer with the table 3 that we have created based on the major comments.

3) Typo errors

There are some typo errors in the text and in tables. For example: Line 37: “rmortality risk”; Table 1: « Ventilated ICU-Patients (N= 3,654)) » « Males N (%)) »

Thank you for your thoughtful look, we have corrected this accordingly.

Reviewer 2 Report

Authors have adressed most of my comments and the quality of the manuscript has significatly improved. However, there are still minor flaws:

-The numbering of the references is wrong - in the introduction section references ranging from 1 to 9 are given and then suddenly no 14. Please correct it.

-Font is still inappropriate - Authors must use Palatino Linotype font  according to the Journal's requirement

Author Response

We would like to thank the reviewers for their suggestions for improvement. We tried to address all the comments and hope that you now find it sufficient for publication. In the following, we will try our best to answer the questions that have been raised.

Changes are highlighted in the manuscript by red color.

Reviewer 1

All my comments and concerns were adequately addressed. I just have a minor comment on the revised manuscript. Some "0" in the table have a different font compared to the other numbers.

Thank you for your thoughtful look, we have corrected this accordingly.

Reviewer 2

Authors have addressed most of my comments and the quality of the manuscript has significantly improved. However, there are still minor flaws:

-The numbering of the references is wrong - in the introduction section references ranging from 1 to 9 are given and then suddenly no 14. Please correct it.

Thank you for your thoughtful look, we have corrected this accordingly.

-Font is still inappropriate - Authors must use Palatino Linotype font according to the Journal's requirement

Thank you for your thoughtful look, we have corrected this accordingly

Reviewer 3

Major comments:

1) Elixhauser Comorbidity Index

Authors used Elixhauser Comorbidity Index as a method to assess comorbidities in COVID-19 affected patients (lines 72-73). But in the current form, it was not detailed further in the manuscript. The index calculation should be presented in the text and in a table in order to know which were the included comorbidities and how this index can be interpreted.

We have included now a table with the phenotypes of the Elixhauser index together with the respective weights for the calculation. Please see table 1 in the manuscript.

2) Patients’ characteristics – times between symptoms onset and admission

Patient characteristics did not include times from symptoms onset to hospital admission and to ICU admission, severity illness score or a risk score of clinical deterioration such as NEWS2. For example, if a large proportion of male patients had a longer time between symptoms onset to hospital admission and were finally admitted in hospital with more severe disease, consequently they may have a higher risk of death or ICU. Then, these variables should be considered since it may introduce bias in results interpretation.

The question is indeed very interesting, unfortunately this data is not included in our dataset. We have written this into the limitations.

3) Patients’ characteristics – Different data in the text and in table 1

Data presentation could be clearer since data presented in the text are different from those reported in the table. In the text, authors reported global data on symptoms at admission (lines 119-125), and in-hospital mortality (lines 131-135) in all patients and in the subgroups, but these data are not presented in tables.

For example, “In total, 21.1% of all patients admitted to a hospital with SARS-CoV-2 died, 47.0% of them had been treated on ICU.” (line 131). These data are not reported in table 1 and not easy to check. “21,1%” can be a percentage of 4,509 intra-hospital deaths (1,152+1,239+817+1,301) on a total of 21,353 patients after exclusion of 8.1%, but it is not clearly presented. All the corresponding data should be presented in the manuscript in a new table or in a new table 1 in order to be clear.

Thank you for your remark. To not be redundant, we intentionally provided information in the text that is not included in the table and vice versa. To make it easier to understand the data at a glance, we have divided the table with the basic characteristics and now there is a detailed list of hospital mortality in five different groups: Total, Non-ICU, ICU, ICU with ventilation, ICU without ventilation. For each of these groups, the total mortality is given (% (n/N)), both for both sexes combined and separately. In addition, ORs and p-values of the comparison between the sexes are given in each case. Below the table is an indication of the proportion of the total data used for the mortality analysis.

Please see table 3 in the manuscript

4) Multivariable analysis and independent risk factors for in hospital mortality in ICU.

Authors used multivariable logistic GLMMs to assess the influence of sex on the risk of admission to the ICU and death in the total cohort (tables 2 and 3).

As a mortality prediction model was further applied to the non-ICU and the ICU cohort, it should be interesting to present multivariate analysis of in hospital mortality in these subgroups in order to known whether age, sex, interaction sex / age, Elixhauser comorbidity index are also independent risk factors.

We have made new calculations which can now be found in an extended table 5

Minor comments:

1) Mortality prediction model in ICU cohort

To compare the risk of death according to age, authors developed a mortality prediction model for the Non-ICU and the ICU cohort (Figure 2). In the statistic section, the model predicting in-hospital mortality was constructed with logistic regression based on sex, age, and the interaction of these variables, with age as a linear and quadratic predictor (second-grade polynomial). As table 1 reported that ICU stay was longer in male than in female patients (10.4 vs 8.0; p<.0001) and that, in ICU, the proportion of ventilated patients was higher in male than in female (65.0% vs 35.0%, p<.0001), it should be interesting to known if these potential explanatory variables have been included in the construction of the mortality prediction model adapted for ICU patients.

This is indeed an interesting question that we would like to answer with the following 2 tables. However, our intention was to develop a model that predicts mortality at the beginning of the stay in the intensive care unit or in the normal ward. To do this, we used the parameters that are easily available at the time of admission. Adding ventilation and intensive care stay would make the model no longer usable for prediction at admission, as these are only available during the course. This would make it more of a retrospective analysis than a prediction model.

Therefore, instead of including the tables in the article, we extended the methods sections with further information on the prediction model.

Table 1: Results of multivariable analyses of in-hospital mortality in the ICU cohort

Variable

OR (95% CI)

P value

Male sex

1.12 (0.97-1.29)

0.1357

Age

1.08 (1.07-1.08)

< 0.0001

Interaction sex × age

1.03 (1.01-1.04)

< 0.0001

Elixhauser comorbidity score

1.04 (1.04-1.05)

< 0.0001

Length of stay at ICU

0.99 (0.98-0.99)

< 0.0001

Mechanical ventilation

11.71 (9.85-13.92)

< 0.0001

* Based on 5,149 cases (84.4%). We excluded cases with discharge due to hospital transfer or unspecified reason.

Table 2: Results of multivariable analyses of in-hospital mortality in the non-ICU cohort.

Variable

OR (95% CI)

P value

Male sex

1.30 (1.08-1.56)

0.0052

Age

1.10 (1.09-1.11)

< 0.0001

Interaction sex × age

1.02 (1.01-1.03)

0.0044

Elixhauser comorbidity score

1.04 (1.03-1.04)

< 0.0001

Mechanical ventilation

11.20 (7.78-16.12)

< 0.0001

* Based on 16,197 cases (94.5%). We excluded cases with discharge due to hospital transfer or unspecified reason.

2) Comparison of in-hospital mortality

Authors compared the in-hospital mortality ICU ventilated patients to all ICU patients and non-ICU patients. It should be interesting to compare in-hospital mortality in non-ICU patients to ICU patients, and ventilated ICU patients to non-ventilated ICU patients, and then focus on sex differences among the different groups.

Thank you for this question, which we answer with the table 3 that we have created based on the major comments.

3) Typo errors

There are some typo errors in the text and in tables. For example: Line 37: “rmortality risk”; Table 1: « Ventilated ICU-Patients (N= 3,654)) » « Males N (%)) »

Thank you for your thoughtful look, we have corrected this accordingly.

Reviewer 3 Report

JCM-1419560-peer-review-v1: Sex Differences in Clinical Course and Intensive Care Unit Admission in a National Cohort of Hospitalized Patients with COVID-19.

This observational retrospective study reported the influence of sex on the risk of admission to the ICU and death in a national cohort which included 23,235 COVID-19 patients from 83 german hospitals between February 4th, 2020 and March 22nd, 2021. Authors showed that males received more intensive care (OR = 1.61) and mechanical ventilation (OR = 1.87) and, thanks to a model for the prediction of mortality, they also showed that the risk of death after 60 years old increased more in males than in females in the non-ICU cohort and more especially in the ICU cohort.

Authors concluded “To our knowledge, this interaction of sex and age on mortality especially among ICU and ventilated patients has not been described before.” However as written in the discussion section, “Male sex and age of patients [4-6] have been identified as risk factors for ICU admission and death in several studies.” Indeed, in their study, Cardoso FS et al (Critical care 2020) included 18,647 patients and reported that age, sex and numbers of comorbidities were independent risk factors for intensive care unit admission or all-cause mortality. In their study, ICU or deceased patients had a median (IQR) age of 80 (69–87) and 54.7% were males whereas in the “no ICU or survived” group the median age was 49 (35–64) and 40.8% were males. So, the reported data presented in this manuscript are interesting but less original.

Major comments:

1) Elixhauser Comorbidity Index 

Authors used Elixhauser Comorbidity Index as a method to assess comorbidities in COVID-19 affected patients (lines 72-73). But in the current form, it was not detailed further in the manuscript. The index calculation should be presented in the text and in a table in order to know which were the included comorbidities and how this index can be interpreted.

2) Patients’ characteristics – times between symptoms onset and admission

Patient characteristics did not include times from symptoms onset to hospital admission and to ICU admission, severity illness score or a risk score of clinical deterioration such as NEWS2. For example, if a large proportion of male patients had a longer time between symptoms onset to hospital admission and were finally admitted in hospital with more severe disease, consequently they may have a higher risk of death or ICU. Then, these variables should be considered since it may introduce bias in results interpretation.

3) Patients’ characteristics – Different data in the text and in table 1

Data presentation could be clearer since data presented in the text are different from those reported in the table. In the text, authors reported global data on symptoms at admission (lines 119-125), and in-hospital mortality (lines 131-135) in all patients and in the subgroups, but these data are not presented in tables. For example, “In total, 21.1% of all patients admitted to a hospital with SARS-CoV-2 died, 47.0% of them had been treated on ICU.” (line 131). These data are not reported in table 1 and not easy to check. “21,1%” can be a percentage of 4,509 intra-hospital deaths (1,152+1,239+817+1,301) on a total of 21,353 patients after exclusion of 8.1%, but it is not clearly presented. All the corresponding data should be presented in the manuscript in a new table or in a new table 1 in order to be clear.

4) Multivariable analysis and independent risk factors for in hospital mortality in ICU.

Authors used multivariable logistic GLMMs to assess the influence of sex on the risk of admission to the ICU and death in the total cohort (tables 2 and 3).

As a mortality prediction model was further applied to the non-ICU and the ICU cohort, it should be interesting to present multivariate analysis of in hospital mortality in these subgroups in order to known whether age, sex, interaction sex / age, Elixhauser comorbidity index are also independent risk factors.

Minor comments:

1) Mortality prediction model in ICU cohort

To compare the risk of death according to age, authors developed a mortality prediction model for the Non-ICU and the ICU cohort (Figure 2). In the statistic section, the model predicting in-hospital mortality was constructed with logistic regression based on sex, age, and the interaction of these variables, with age as a linear and quadratic predictor (second-grade polynomial). As table 1 reported that ICU stay was longer in male than in female patients (10.4 vs 8.0; p<.0001) and that, in ICU, the proportion of ventilated patients was higher in male than in female (65.0% vs 35.0%, p<.0001), it should be interesting to known if these potential explanatory variables have been included in the construction of the mortality prediction model adapted for ICU patients.

2) Comparison of in-hospital mortality

Authors compared the in-hospital mortality ICU ventilated patients to all ICU patients and non-ICU patients. It should be interesting to compare in-hospital mortality in non-ICU patients to ICU patients, and ventilated ICU patients to non-ventilated ICU patients, and then focus on sex differences among the different groups.

3) Typo errors

There are some typo errors in the text and in tables. For example: Line 37: “rmortality risk”; Table 1: « Ventilated ICU-Patients (N= 3,654)) » « Males N (%)) »

Author Response

We would like to thank the reviewers for their suggestions for improvement. We tried to address all the comments and hope that you now find it sufficient for publication. In the following, we will try our best to answer the questions that have been raised.

Changes are highlighted in the manuscript by red color.

Reviewer 1

All my comments and concerns were adequately addressed. I just have a minor comment on the revised manuscript. Some "0" in the table have a different font compared to the other numbers.

Thank you for your thoughtful look, we have corrected this accordingly.

Reviewer 2

Authors have addressed most of my comments and the quality of the manuscript has significantly improved. However, there are still minor flaws:

-The numbering of the references is wrong - in the introduction section references ranging from 1 to 9 are given and then suddenly no 14. Please correct it.

Thank you for your thoughtful look, we have corrected this accordingly.

-Font is still inappropriate - Authors must use Palatino Linotype font according to the Journal's requirement

Thank you for your thoughtful look, we have corrected this accordingly

Reviewer 3

Major comments:

1) Elixhauser Comorbidity Index

Authors used Elixhauser Comorbidity Index as a method to assess comorbidities in COVID-19 affected patients (lines 72-73). But in the current form, it was not detailed further in the manuscript. The index calculation should be presented in the text and in a table in order to know which were the included comorbidities and how this index can be interpreted.

We have included now a table with the phenotypes of the Elixhauser index together with the respective weights for the calculation. Please see table 1 in the manuscript.

2) Patients’ characteristics – times between symptoms onset and admission

Patient characteristics did not include times from symptoms onset to hospital admission and to ICU admission, severity illness score or a risk score of clinical deterioration such as NEWS2. For example, if a large proportion of male patients had a longer time between symptoms onset to hospital admission and were finally admitted in hospital with more severe disease, consequently they may have a higher risk of death or ICU. Then, these variables should be considered since it may introduce bias in results interpretation.

The question is indeed very interesting, unfortunately this data is not included in our dataset. We have written this into the limitations.

3) Patients’ characteristics – Different data in the text and in table 1

Data presentation could be clearer since data presented in the text are different from those reported in the table. In the text, authors reported global data on symptoms at admission (lines 119-125), and in-hospital mortality (lines 131-135) in all patients and in the subgroups, but these data are not presented in tables.

For example, “In total, 21.1% of all patients admitted to a hospital with SARS-CoV-2 died, 47.0% of them had been treated on ICU.” (line 131). These data are not reported in table 1 and not easy to check. “21,1%” can be a percentage of 4,509 intra-hospital deaths (1,152+1,239+817+1,301) on a total of 21,353 patients after exclusion of 8.1%, but it is not clearly presented. All the corresponding data should be presented in the manuscript in a new table or in a new table 1 in order to be clear.

Thank you for your remark. To not be redundant, we intentionally provided information in the text that is not included in the table and vice versa. To make it easier to understand the data at a glance, we have divided the table with the basic characteristics and now there is a detailed list of hospital mortality in five different groups: Total, Non-ICU, ICU, ICU with ventilation, ICU without ventilation. For each of these groups, the total mortality is given (% (n/N)), both for both sexes combined and separately. In addition, ORs and p-values of the comparison between the sexes are given in each case. Below the table is an indication of the proportion of the total data used for the mortality analysis.

Please see table 3 in the manuscript

4) Multivariable analysis and independent risk factors for in hospital mortality in ICU.

Authors used multivariable logistic GLMMs to assess the influence of sex on the risk of admission to the ICU and death in the total cohort (tables 2 and 3).

As a mortality prediction model was further applied to the non-ICU and the ICU cohort, it should be interesting to present multivariate analysis of in hospital mortality in these subgroups in order to known whether age, sex, interaction sex / age, Elixhauser comorbidity index are also independent risk factors.

We have made new calculations which can now be found in an extended table 5

Minor comments:

1) Mortality prediction model in ICU cohort

To compare the risk of death according to age, authors developed a mortality prediction model for the Non-ICU and the ICU cohort (Figure 2). In the statistic section, the model predicting in-hospital mortality was constructed with logistic regression based on sex, age, and the interaction of these variables, with age as a linear and quadratic predictor (second-grade polynomial). As table 1 reported that ICU stay was longer in male than in female patients (10.4 vs 8.0; p<.0001) and that, in ICU, the proportion of ventilated patients was higher in male than in female (65.0% vs 35.0%, p<.0001), it should be interesting to known if these potential explanatory variables have been included in the construction of the mortality prediction model adapted for ICU patients.

This is indeed an interesting question that we would like to answer with the following 2 tables. However, our intention was to develop a model that predicts mortality at the beginning of the stay in the intensive care unit or in the normal ward. To do this, we used the parameters that are easily available at the time of admission. Adding ventilation and intensive care stay would make the model no longer usable for prediction at admission, as these are only available during the course. This would make it more of a retrospective analysis than a prediction model.

Therefore, instead of including the tables in the article, we extended the methods sections with further information on the prediction model

Table 1: Results of multivariable analyses of in-hospital mortality in the ICU cohort

Variable

OR (95% CI)

P value

Male sex

1.12 (0.97-1.29)

0.1357

Age

1.08 (1.07-1.08)

< 0.0001

Interaction sex × age

1.03 (1.01-1.04)

< 0.0001

Elixhauser comorbidity score

1.04 (1.04-1.05)

< 0.0001

Length of stay at ICU

0.99 (0.98-0.99)

< 0.0001

Mechanical ventilation

11.71 (9.85-13.92)

< 0.0001

* Based on 5,149 cases (84.4%). We excluded cases with discharge due to hospital transfer or unspecified reason.

Table 2: Results of multivariable analyses of in-hospital mortality in the non-ICU cohort.

Variable

OR (95% CI)

P value

Male sex

1.30 (1.08-1.56)

0.0052

Age

1.10 (1.09-1.11)

< 0.0001

Interaction sex × age

1.02 (1.01-1.03)

0.0044

Elixhauser comorbidity score

1.04 (1.03-1.04)

< 0.0001

Mechanical ventilation

11.20 (7.78-16.12)

< 0.0001

* Based on 16,197 cases (94.5%). We excluded cases with discharge due to hospital transfer or unspecified reason.

2) Comparison of in-hospital mortality

Authors compared the in-hospital mortality ICU ventilated patients to all ICU patients and non-ICU patients. It should be interesting to compare in-hospital mortality in non-ICU patients to ICU patients, and ventilated ICU patients to non-ventilated ICU patients, and then focus on sex differences among the different groups.

Thank you for this question, which we answer with the table 3 that we have created based on the major comments.

3) Typo errors

There are some typo errors in the text and in tables. For example: Line 37: “rmortality risk”; Table 1: « Ventilated ICU-Patients (N= 3,654)) » « Males N (%)) »

Thank you for your thoughtful look, we have corrected this accordingly.

This manuscript is a resubmission of an earlier submission. The following is a list of the peer review reports and author responses from that submission.

Round 1

Reviewer 1 Report

The paper presents the results of a retrospective observational study of the sex differences in the clinical course of >20000 hospitalized patients with COVID-19. The study supports a sex bias of adverse outcome of COVID-19 with increased age. 

The methods and results are logically described and discussed. The analysis however does not take into account potential differences in medication or laboratory parameters. It would interesting to study and discuss potential contribution of these parameters in the multivariate analysis. 

The authors report the overall results of patients treated in 83 hospitals. Does this reflect the results in all/most hospitals? Could the sex differences observed reflect the results in one or few hospitals with many patients that skew the data? 

The authors should expand a bit the introduction and discussion sections.

Author Response

We would like to thank the editor and the reviewers for considering our manuscript for publication. We tried to address all the comments and would like to thank the reviewers and the editor for their thorough and attentive review that was very helpful in bringing the article forward. In the following, we will try our best to answer the questions that have been raised and the article has now been revised by a native speaking medical colleague.

Changes are highlighted in the manuscript by red colour.

The methods and results are logically described and discussed. The analysis however does not take into account potential differences in medication or laboratory parameters. It would interesting to study and discuss potential contribution of these parameters in the multivariate analysis.

Thank you for your remark. It would indeed be interesting to analyze the contribution of medication and laboratory parameters. Our study however was conducted as a retrospective analysis of claims data and data from a routine infection control system; which did not provide information on lab results or medication.

The authors report the overall results of patients treated in 83 hospitals. Does this reflect the results in all/most hospitals?

Thank you for your remark. The following sentence has been added to the methods section:

The proportion of basic to tertiary care is representative for the overall distribution of hospitals in Germany. Also, the patient mix is representative, since all Helios hospitals are fully covered by all health care insurance plans.

Could the sex differences observed reflect the results in one or few hospitals with many patients that skew the data?

We analyzed the data with (generalized) linear mixed models with hospitals as random factor. These models take into account the variability of the hospitals with respect to the dependent variables. Observations of institutions with very low or high values are shrunken towards the mean and hence the corresponding cases enter the analyses with lower weight.

The authors should expand a bit the introduction and discussion sections.

We have expanded the introduction and the discussion according to the recommendations of the reviewers

Reviewer 2 Report

General

This manuscript by Nachtigall et al. describes gender-dependent differences in clinical course of hospitalized patients with COVID-19. The topic is up-to-date, paper provides some useful information about these disparities and the study sample is large.  However, there are some problems. I would also suggest to have the manuscript proofread by a professional correcting agency, as there are some grammatical errors, as well as some sentences are unclear and hard to follow.

Abstract

  • Line 31 – the correct sentence will be: “The aim of the study…”.
  • Line 32 – In my opinion the aim of the study is not clear for the readers and it should be clarified.
  • There are some language mistakes, for example in line 36 “Male sex accounted for 51.4%.” I should be changed into: “Male sex accounted for 51.4% of total sample”.
  • Line 41 – if authors use any abbreviation (ICU) it should be previously explained.
  • At the end of the abstract a short sentence summarizing the whole paper or indicating practical implications of this study must be added.

Introduction

  • In fact, the introduction section is really scarce and it does not provide an appropriate background in terms of presented topic.
  • It will be of great interest to expand an explanation about possible factors which make men more vulnerable group to severe course or death due to COVID, comparing with women. Actually there is very little information on this issue.
  • Line 51 – Authors say that: “Age has also been identified as risk factor for an adverse outcome [7].” – does literature provides any information on what age is considered to be risk factor for adverse outcomes?
  • Line 52 – Authors say that: “Little 51 is known however on the interaction between sex and age on mortality”. Are there any studies which covered this topic? If so, Authors should cite them.
  • Line 55 – the correct sentence will be: “The aim of the study…”.

Materials and Methods

  • Important information on studied population is missing. Specifically, it should be described what was the sampling procedure, what was the total number of participants
  • There is no information if Authors calculated sample size – please add it. What was your supposed minimum/ maximum number of participants included to the study?
  • Can Authors explain why data on COVID-19 symptoms at admission were not available for all patients?
  • Line 92 – Authors must briefly say what is Elixhauser comorbidity index,

Results

  • Results are not presented in a way which is required by Journal of Clinical Medicine. Please modify in all tables font, font size
  • Please report p-value up to four decimal points

Discussion

  • Authors must expand their discussion section, as little is known whether there were similar studies to this German one, which analyzed gender-dependent differences in the course of COVID-19. If so, Authors must present them.

Conclusion

  • This section is comprehensive and highlights the most important information from the article.

Author Response

We would like to thank the editor and the reviewers for considering our manuscript for publication. We tried to address all the comments and would like to thank the reviewers and the editor for their thorough and attentive review that was very helpful in bringing the article forward. In the following, we will try our best to answer the questions that have been raised and the article has now been revised by a native speaking medical colleague.

Changes are highlighted in the manuscript by red colour.

Abstract

Line 31 – the correct sentence will be: “The aim of the study…”.

The correction has been done

Line 32 – In my opinion the aim of the study is not clear for the readers and it should be clarified.

Thank you for this remark, the aim has been rephrased.

There are some language mistakes, for example in line 36 “Male sex accounted for 51.4%.” I should be changed into: “Male sex accounted for 51.4% of total sample”.

Thank you for your remark. A native speaker has now revised the article.

Line 41 – if authors use any abbreviation (ICU) it should be previously explained.

We have added an explanation of ICU in the abstract

At the end of the abstract a short sentence summarizing the whole paper or indicating practical implications of this study must be added.

Thank you for your remark. We added the following sentence to the abstract:

In conclusion, the sex gap was most evident among the oldest in ICU. Understanding sex-determined differences in COVID-19 can be useful to facilitate an individualized treatment.

Introduction

In fact, the introduction section is really scarce and it does not provide an appropriate background in terms of presented topic.

The introduction has been rewritten providing more details on the background.

It will be of great interest to expand an explanation about possible factors which make men more vulnerable group to severe course or death due to COVID, comparing with women. Actually there is very little information on this issue.

The introduction has been rewritten providing more information on the known physiological differences responsible for the worse outcome in men.

Line 51 – Authors say that: “Age has also been identified as risk factor for an adverse outcome [7].” – does literature provides any information on what age is considered to be risk factor for adverse outcomes?

We apologize for the misleading wording, to make this statement more understandable, age has been replaced by older age.

Line 52 – Authors say that: “Little 51 is known however on the interaction between sex and age on mortality”. Are there any studies which covered this topic? If so, Authors should cite them.

In the course of the revision of the introduction, this sentence was removed

Line 55 – the correct sentence will be: “The aim of the study…”.

In the course of the revision of the introduction, this sentence was removed

Materials and Methods

Important information on studied population is missing. Specifically, it should be described what was the sampling procedure, what was the total number of participants There is no information if Authors calculated sample size – please add it. What was your supposed minimum/ maximum number of participants included to the study?

The study was conducted as an observational retrospective cohort study. All patients hospitalized with SARS-CoV2 in the study period were included. Due to the study design, no sample size was calculated.

Can Authors explain why data on COVID-19 symptoms at admission were not available for all patients?

The part of the symptoms comes from a manually managed database, there was no automatic transmission of findings in this database, so that it was not automatically included in the routine database for every patient. We have specified the methodology part for this.

Line 92 – Authors must briefly say what is Elixhauser comorbidity index,

We have included a short sentence explaining the Elixhauser comorbidity index.

Results

Results are not presented in a way which is required by Journal of Clinical Medicine. Please modify in all tables font, font size

Thank you for the careful observation, we have adjusted the font

Please report p-value up to four decimal points

We now report four decimal points for all p-values.

Discussion

Authors must expand their discussion section, as little is known whether there were similar studies to this German one, which analyzed gender-dependent differences in the course of COVID-19. If so, Authors must present them.

We have expanded the introduction and the discussion according to the recommendations of the reviewers.

Conclusion

This section is comprehensive and highlights the most important i